# An Exploratory Study on Spoilage Bacteria and *Listeria monocytogenes* in Fresh Salmon: Extending Shelf-Life Using Vacuum and Seasonings as Natural Preservatives

**DOI:** 10.3390/vetsci10070423

**Published:** 2023-06-29

**Authors:** Maria-Leonor Lemos, Joana C. Prata, Inês C. Rodrigues, Sofia Martins-Costa, Bernardo Archer, Jorge Machado, Rui Dilão, Paulo Vaz-Pires, Paulo Martins da Costa

**Affiliations:** 1ICBAS—Instituto de Ciências Biomédicas Abel Salazar, Universidade do Porto, Rua de Jorge Viterbo Ferreira, 228, 4050-313 Porto, Portugal; mllemos@icbas.up.pt (M.-L.L.); joana.prata@iucs.cespu.pt (J.C.P.); icrodrigues@icbas.up.pt (I.C.R.); bernarcher@gmail.com (B.A.); jmachado@icbas.up.pt (J.M.); vazpires@icbas.up.pt (P.V.-P.); 2CIIMAR—Interdisciplinary Centre of Marine and Environmental Research, Terminal de Cruzeiros do Porto de Leixões, Av. General Norton de Matos s/n, 4450-208 Matosinhos, Portugal; 3TOXRUN—Toxicology Research Unit, University Institute of Health Sciences—CESPU (IUCS−CESPU), 4585-116 Gandra, Portugal; 4Department of Physics, Instituto Superior Técnico, University of Lisbon, Av. Rovisco Pais, 1049-001 Lisbon, Portugal; sofiadcosta@tecnico.ulisboa.pt (S.M.-C.); ruidilao@tecnico.ulisboa.pt (R.D.); 5Soguima—Comércio e Indústria Alimentar S.A., Zona Industrial II, 4805-559 Guimarães, Portugal

**Keywords:** salmon, shelf-life, spoilage microbiota, *Listeria monocytogenes*, biopreservation

## Abstract

**Simple Summary:**

The study aims at improving the shelf-life and safety of fresh salmon, a highly perishable food often consumed raw in sushi or sashimi. Good hygiene practices and vacuum packaging extended shelf-life up to 20 days. The presence of the bacteria *Carnobacterium maltaromaticum* inhibited *Listeria monocytogenes* growth, which may be responsible for foodborne illnesses. Seasonings, namely oregano oil, lemon juice, garlic powder, and salt (NaCl), showed inhibitory effects on spoilage bacteria and *L. monocytogenes*. The seasonings’ mixtures also exhibited bacteriostatic and bactericidal effects. Finally, a medium produced from salmon was developed and it showed a good correlation with bacterial growth in a standard commercial culture medium.

**Abstract:**

A growing population increases the demand for food, but short shelf-lives and microbial hazards reduce supply and increase food waste. Fresh fish is highly perishable and may be consumed raw, such as salmon in sushi. This work aims to identify strategies to improve the shelf-life and safety of fresh salmon, using available methods (i.e., vacuum) and exploring the use of natural preservatives (i.e., seasonings). Vacuum packaging and good hygiene practices (which reduce initial flora) extended shelf-life up to 20 days. *Carnobacterium maltaromaticum* was dominant in vacuum packaging conditions and showed potential for inhibiting *Listeria monocytogenes*. For natural preservatives, *L. monocytogenes* required higher inhibitory concentrations in vitro when compared to the 10 spoilage bacteria isolated from fresh salmon fillets, presenting a minimum inhibitory concentration (MIC) of 0.13% for oregano essential oil (OEO), 10% for lemon juice, 50 mg mL^−1^ for garlic powder, and >10% for NaCl. A good bacteriostatic and bactericidal effect was observed for a mixture containing 5% NaCl, 0.002% OEO, 2.5% lemon juice, and 0.08 mg mL^−1^ garlic powder. Finally, using the salmon medium showed an adequate correlation with the commercial culture medium.

## 1. Introduction

Fish consumption varies widely across countries, for instance, from 6–60 kg per person per year in different member states of the European Union [1]. Besides cultural differences, the consumption of fresh fish is often limited by its short shelf-life (e.g., within 48 h of purchase when kept under refrigeration [2]). A shorter shelf-life leads to losses as food waste, which corresponds annually to 10 million tons or 3–8% of fish products [3]. Fish spoilage occurs due to autolysis, lipid oxidation, and microbial spoilage which also releases unpleasant compounds (e.g., putrescine) [4]. Moreover, microbial spoilage considerably impacts food safety and security, the environment, and the economy [5,6,7]. Thus, the fish-processing industry is actively seeking alternative methods of shelf-life extension and marketability of fresh fish. Besides refrigeration, methods to delay spoilage and improve food safety mainly rely on heat treatment and the use of synthetic preservatives [8]. However, these treatments may trigger unpleasant sensory changes and are unsatisfactory to consumers seeking less-processed food [9,10]. In addition, food synthetic preservatives have raised public health concerns as potential drivers for resistance to clinically relevant antibiotics among commensal and foodborne bacteria [11]. Another traditional method of shelf-life extension is the use of vacuum or modified atmosphere packaging for controlling microbiological growth, which may also reduce the sensorial quality of fish [12].

Natural preservatives provide an alternative that increases shelf-life and food safety while possibly enhancing food’s organoleptic characteristics [9,13]. The effectiveness of natural preservatives on spoilage bacteria has been previously documented, but extrapolating these results from in vitro to industrial settings remains a challenge [8,14] since its inhibitory activity may be limited by the interaction with the food matrix [7]. Moreover, natural preservatives may present antimicrobial activity against foodborne pathogens. For instance, *Listeria* spp. isolated from fish was inhibited by natural preservatives, such as 1.25% ascorbic acid and 0.2% oregano oil [15]. *Listeria* spp. presents a problem to the food industry since it can be introduced into the product through raw materials or cross-contamination, grows under refrigeration, and exposure to low infective doses may cause disease with high hospitalization and high mortality rates [16]. A recent outbreak related to cold-smoked or gravad salmon consumption resulted in 22 listeriosis cases and 5 deaths in the European Union [17]. Food poisoning may also result from consuming fresh fish, such as raw salmon in sushi and sashimi [18]. Therefore, fresh salmon is a main candidate for biopreservation, that is, preservation using natural microorganisms and antimicrobials. So far, limited research has been conducted on fresh salmon preservation compared to smoked salmon, especially addressing novel preservation strategies (e.g., biopreservatives). Thus, this exploratory work aimed at identifying possible strategies to extend the shelf-life and microbiological safety of fresh salmon, measuring the capacity for microbial inhibition of an available and well-recognized method (vacuum packaging) and of an emerging strategy using natural preservatives (seasonings).

## 2. Materials and Methods

### 2.1. Preparation of Salmon Fillets

Two batches of farmed Atlantic salmon (*Salmo salar*; weight: 4 ± 0.5 kg) were harvested in Norway from an offshore fish farming facility and were provided by a Portuguese seafood company (Pedro & Peixe, Lda., Matosinhos, Portugal) between February and April in 2022. Salmon was obtained within 72–96 h of harvest and transported on ice in expanded polystyrene (EPS) boxes to the laboratory (Laboratory of Food Technology and Microbiology, ICBAS, University of Porto) within 2 h. Upon arrival, the fish were immediately scaled, headed, filleted, and cut into rectangular slices (100 ± 10 g).

Salmon fillets were rinsed with salted tap water (3% NaCl), following the costumery procedure of the local industry, followed by aerobic and vacuum packaging. The vacuum-packed fillets were packed in polyamide and polyethylene bags (Termofilm, Bente, Portugal) and sealed using a single-chamber vacuum-sealing unit (Webomatic easy PACK-mk2, Bochum, Germany). The final vacuum pressure used was 25 mbar, obtaining a degree of filling of approximately 33%. Then, all fillets (under aerobic and vacuum conditions) were stored at 4 °C for 20 days.

### 2.2. Microbiological Analysis

#### 2.2.1. Microbiota Dynamics in Salmon Fillets during Storage

The microbiota dynamics of salmon fillets were assessed by collecting samples at four-day intervals until reaching 10^8^ colony-forming units (CFUs) per gram or 20 days of storage. The total aerobic colony enumeration complied with the international standard ISO 4833:2003. Briefly, each group’s representative sample (10 g) was aseptically weighed, transferred to a sterile stomacher bag, and homogeneously suspended in 1/10 buffered peptone water (BPW, Biokar Diagnostics, Allonne, France). Then, 1 mL of tenfold serial dilutions were inoculated by incorporation into plate count agar (PCA, Biokar Diagnostics) and incubated at 30 °C for 72 h, according to ISO 4833:2003. The protocol followed was similar for psychrophilic microorganisms, but the plates were incubated at 7 °C for 7 days.

Four colonies were selected for microbiota characterization based on colony size from the higher dilution at each sampling time. Gram-positive and Gram-negative bacteria were distinguished by Gram coloration and biochemical tests (namely catalase and oxidase tests). Representative isolates of each group and sampling moment were selected for taxonomic identification by 16S ribosomal rRNA gene analysis. After PCR amplification using the 27f and 1492r primers [19], 16S rDNA products were purified and sequenced by Eurofins Genomics, and the resulting sequences were analyzed in the BLAST network services at the NCBI—National Center for Biotechnology Information, Bethesda, MA, USA.

From 50 bacterial species identified by 16S rDNA (Appendix A, Table A1), 10 isolates were selected for further studies based on the following criteria: (i) belonging to different bacterial species (diversity criterion); (ii) presence of bacterial species in both assays (inclusion criterion); (iii) frequency of the species in each of the assays (first tiebreaker); (iv) relevance in the salmon degradation process according to the literature (second tiebreaker) [18,20]. Given their relevance in terms of food safety, including for salmon fillets as a source of cross-contamination or consumption of raw preparations (e.g., sushi) [21], *Listeria monocytogenes* 12MOB099LM [22] and a food strain isolated from smoked salmon were provided by the Portuguese National Health Institute (INSA) and added to the group of bacteria under study.

#### 2.2.2. Co-Growth of *Listeria monocytogenes* with Salmon Fillet Microbiota

Starting with a solution of 10^8^ CFU/g, each strain of *Listeria monocytogenes* (*L. monocytogenes*) was paired with each one of the 10 degradative bacteria at a 1:1 ratio in a 96-well plate and incubated for 24 h at 28 °C. The growth rate was assessed by plate count on Muller–Hinton agar (MHA, Biokar Diagnostics) based on bacterial colony size and morphology differences.

### 2.3. In Vitro Antimicrobial Effect of Natural Preservatives (Seasonings) and Mixtures

Four seasonings frequently used in Mediterranean cuisine were selected considering their inhibitory effect over major spoilage and foodborne bacteria of raw salmon [5,9] and tested on the 10 bacteria isolates previously identified in salmon and on the two *L. monocytogenes* strains. Selected seasonings were consecutively diluted by a factor of two in the growth medium to achieve the tested concentrations: (i) sodium chloride (NaCl; 2.5–10%) (Formulab, Matosinhos, Portugal); (ii) oregano essential oil (OEO; 1–0.002%) (Biover, Nazareth, Belgium); (iii) lemon juice (10–0.625%) (Modelo Continente Hipermercados, Matosinhos, Portugal); and (iv) garlic powder (50–0.08 mg mL^−1^) (Margão, Sobralinho, Portugal).

#### 2.3.1. Susceptibility Testing

The minimal inhibitory concentration (MIC) of each natural preservative was determined by broth microdilution method as described by the Clinical and Laboratory Standard Institute guidelines [23]. Two-fold serial dilutions of the seasonings were prepared in a 96-well plate using cation-adjusted Mueller–Hinton broth (CAMHB, Sigma-Aldrich, St. Louis, MO, USA). All bacterial strains were cultured in MHA (Biokar Diagnostics) and incubated overnight prior to each MIC determination assay. Then, the optical density at 600 nm of bacterial suspension was measured and adjusted to 0.1. The suspension was further diluted in order to achieve a final inoculum of 5 × 10^5^ CFU mL^−1^ in each well. The microplates were then incubated at 28 °C for 48 h. The MIC was determined by visual inspection as the lowest concentration with no visible growth. Each assay included a negative control (with no inoculum) and a positive control (with no seasoning) and monitoring of inoculum density by colony counting. To assess the minimal bactericidal concentration (MBC), 10 µL of culture were collected from wells that showed no visible growth and were spread on MHA plates. The MBC was determined as the lowest concentration at which no colonies grew after incubation for 24 h at 28 °C.

#### 2.3.2. Inhibitory Potential of Mixtures of Natural Preservatives (Seasonings)

Based on the MIC values, seasoning mixtures were formulated based on different criteria (Appendix A, Table A2). Mixture 1 (5% NaCl, 0.002% OEO, 2.5% lemon juice, 0.08 mg mL^−1^ garlic powder) aimed at containing at least the MIC of all the 10 bacteria isolates found to be dominant during storage. Mixture 2 (2.5% NaCl, 0.002% OEC, 0.625% lemon juice, 3.13 mg mL^−1^ garlic powder) was selected based on the MIC of the *Listeria monocytogenes* strains. Mixture 3 (0.25% NaCl, 0.005% OEO, 0.25% lemon juice, 0.06 mg mL^−1^ garlic powder) was based on in vivo experiments from the literature [5], having as an adjustment to the expected marinade absorption rate of 2.5%, obtained experimentally in salmon fillets (data not shown).

To assess the in vitro inhibitory activity of each mixture on the set of bacteria, each strain was suspended in CAMHB (Sigma-Aldrich), and the optical density at 600 nm was measured and adjusted to 0.1. A pool with equal proportions of each bacterial inoculum was further diluted to obtain a final concentration of 5 × 10^5^ CFU mL^−1^ in a total volume of 2.5 mL. The bacterial suspensions (set of bacteria with each mixture) were then incubated at 28 °C. Enumeration of total aerobic microorganisms was performed at 0, 2, 4, 8, and 16 days of incubation by plating 100 µL of the bacterial suspensions onto plate count agar (PCA, Biokar Diagnostics) at 28 °C. Additionally, 100 µL of the bacterial suspensions were also inoculated on six selective media in order to distinguish bacterial species and incubated at 28 °C: Kanamycin Aesculin Azide (KAA, Oxoid, Hampshire, United Kingdom) agar allowing the growth of the *Pseudomonas* species; Hektoen Enteric Agar (HEA, Biokar Diagnostics) for enumeration of *Carnobacterium maltaromaticum* (*C. maltaromaticum*); cephaloridine fucidin cetrimide (CFC, Biokar Diagnostics) agar for the growth of *Pseudomonas gessardi* (*P. gessardi*), *Pseudomonas psychrophila* (*P. psychrophila*), *Pseudomonas fragi* (*P. fragi*), *Psychrobacter cibarius* (*P. cibarius*), *Pseudomonas jessini* (*P. jessini*), and *Psychrobacter maritimus* (*P. maritimus*); Aeromonas Medium base (RYAN, Oxoid) on which *P. gessardi*, *P. psychrophila*, *P. fragi*, *P. jessini*, *Shewanella putrefaciens* (*S. putrefaciens*), and *P. maritimus* could be counted; Compass Listeria agar (Biokar Diagnostics) for the differentiation of both *L. monocytogenes* strains; CHROMagar Orientation (CHROMagar, Paris, France) allowing the growth of *Brochothrix thermosphacta* (*B. thermosphacta*), *C. maltaromaticum*, and *L. monocytogenes*; and Triple Sugar Iron agar (TSI Agar, Oxoid) for the counting of *Shewanella baltica* (*S. Baltica*) and *S. putrefaciens*.

### 2.4. Comparative Growth Rate between Commercial Culture Medium and Food Matrix Media

To replicate the conditions found in the salmon fillet, a study compared the kinetic growth rate of a reference culture medium CAMHB (Sigma-Aldrich) with a salmon medium, that was prepared by adding macerated salmon fillets (16 g) to deionized water (100 mL), followed by filtration of the suspension using 0.45 μm filter to remove microorganisms originally present in the salmon. The growth rate of the ten isolates at 28 °C was evaluated by optical density measurements, and CFU counts at 0, 3, 6, 12, and 24 h, in two independent assays.

### 2.5. Statistical Analysis

Maximum slope (µ) and lag phase duration (LAG) were calculated based on Baranyi models, using Microrisk Lab v1.2 ([24]). Wilcoxon signed-rank tests were conducted to compare the parameter for each species under two different conditions in IBM SPSS Statistics version 26, considering α = 0.05. Results are expressed as base 10 logarithm (log).

## 3. Results

### 3.1. Characterization of Bacterial Dynamics in Salmon during Storage

A first trial assessed the microbiological quality of two salmon fillets under aerobic and vacuum-packed conditions at 4 °C. The number of total microorganisms was 0–1 log CFU g^−1^ on day 0. Two samples stored under aerobic conditions reached 8.6 and 7.1 log CFU g^−1^ on day 12. Conversely, storage under anaerobic conditions (vacuum) translated into 4.6 and 3.7 log CFU g^−1^ on day 20 (Appendix A, Figure A1). Therefore, a reduction of at least 3.3 log CFU g^−1^ on day 20 was observed for the vacuum packaging of salmon fillets.

Clear differences were found for predominant bacterial species between conditions (Appendix A, Table A3), with a prevalence of *Pseudomonas* spp. and *Psychrobacter* spp. in aerobic conditions, and of Gram-positives and particularly of *C. maltaromaticum* in anaerobic conditions. Based on the identified species and tiebreaker (previously detailed), the following ten isolates were selected for further testing: (i) *P. gessardi;* (ii) *P. psychrophila*; (iii) *P. fragi*; (iv) *P. cibarius*; (v) *B. thermosphacta*; (vi) *C. maltaromaticum*; vii) *P. jessini;* (viii) *S. baltica*; (ix) *S. putrefaciens*; and (x) *P. maritimus*. *L. monocytogenes* 12MOB099LM and a food strain isolated from smoked salmon were also considered.

Co-culture with the previously mentioned bacteria reveals an increase in yield of *L. monocytogenes* strains at 24 h in the presence of most insolates, including *P. cibarius, P. jessini, S. baltica, S. putrefaciens* (>0.2 log CFU g^−1^ at 24 h) and a decrease in the presence of *C. maltaromaticum* (<0.2 log CFU g^−1^ at 24 h) (Figure 1).

### 3.2. In Vitro Antimicrobial Effect of Different Natural Preservatives (Seasonings) and Mixtures

A second trial was conducted to test the use of seasoning to increase the shelf-life of salmon fillets. MIC and MBC of NaCl, OEO, lemon juice, and garlic powder were assessed using a commercial culture medium (CAMHB) and are presented in Table 1. In summary, OEC exhibited a broad spectrum of activity against all tested bacteria, while *L. monocytogenes* generally presented higher MIC and MBC than spoilage bacteria.

Based on MIC results, three seasoning mixtures were prepared using different concentrations (Figure 2). Mixture 1 inhibited total aerobic microorganisms at 28 °C during the full test period, corresponding to a 7.9 log CFU g^−1^ reduction at 16 h (Figure 2a). Mixture 2 and Mixture 3 had lower inhibitory activities (1.70 and 0.66 log CFU g^−1^ at 16 h, respectively), which likely results from the lower concentrations of NaCl and lemon when compared to Mixture 1 (Figure 2a). Regarding specific groups of bacteria, Gram-positive bacteria presented an inhibition trend similar to the number of total microorganisms but with complete inhibition by Mixture 1 (Figure 2b). *Pseudomonas* spp. was inhibited by Mixture 1 and Mixture 2 (reduction of 7.4 and 6.5 log CFU g^−1^ at 16 h, respectively; Figure 2c) and both *L. monocytogenes* strains were completely inhibited by Mixture 1 (Figure 2d).

### 3.3. Comparative Growth Rate between Commercial and Salmon Media

Parameters of growth curves were calculated using a Baranyi model based on results for CFU in the two replicates conducted for two conditions (commercial culture and salmon media) for 10 predominant bacterial isolates to better simulate conditions in the fillet during storage. Four bacterial species were excluded from statistical analysis due to the presence of outliers. The analysis of the remaining six bacterial species showed no statistical difference between the commercial culture and salmon media for the maximum slope (*p* = 0.463; 0.745 vs. 0.740), lag phase (*p* = 0.116; 2.06 vs. 3.73 h), and initial concentration (*p* = 0.075; 5.860 vs. 5.675 log CFU g^−1^). Statistically significant differences were found for the maximum population density, with a reduction of 0.6 log cycle in the salmon media (*p* = 0.046; 9.180 vs. 8.475 log CFU g^−1^). Moreover, the Spearman correlation matrix suggested that 24 h results were strongly correlated for CFU between commercial culture media and salmon media (*p* ≤ 0.020; r_s_ ≥ 0.827).

## 4. Discussion

The microbial spoilage of fish has been defined by the acceptability limit of 7.0 log CFU g^−1^ for mesophilic bacteria, as proposed by the International Commission on Microbiological Specification for Foods and supported by studies on organoleptic losses and decomposition [25,26]. The study of fresh chilled salmon is particularly interesting due to its value, high volume, and consumption as a raw product (e.g., sushi, sashimi) [18]. In the present work, fresh salmon stored at 4 °C exceeded acceptability limits on days 8–12, whereas packaging in a vacuum at 4 °C reduced bacterial growth to < 5 log CFU g^−1^ for up to 20 days. The shelf-life of fresh salmon has been previously estimated as 5–15 days [27]. Conversely, storage under vacuum at 5 °C has reached threshold values only after 6–7 days, which likely results from a higher initial flora (4 log CFU g^−1^) [28] compared to the present study (0–1 log CFU g^−1^). While organoleptic analysis is out of the scope of this manuscript, a previous study has noted that the shelf-life of vacuum-packaged fresh salmon may be shorter due to sensorial changes that were detectable by an expert panel on day 7 [29]. Therefore, the shelf-life of fresh salmon can be extended by implementing good hygiene practices and by storing fillets at 4 °C under a vacuum, but more research is needed to consider changes in organoleptic properties.

The anaerobic conditions of the vacuum packaging led to a shift in predominant species from *Pseudomonas* spp. and *Psychrobacter* spp. to *C. maltaromaticum.* While *Photobacterium phosphoreum* is commonly found in vacuum-packaged fish [18], it was not detected in the present study. Lactic acid bacteria, including *Carnobacterium* spp., have previously been reported in vacuum-packaged salmon [29]. Indeed, *C. maltaromaticum* is a lactic acid bacteria that thrive under aerobic conditions and low temperatures, known for inhibiting pathogenic and degradative microorganisms while having a variable spoilage activity depending on the matrix [30]. Moreover, *Carnobacterium* spp. has a bacteriostatic and bactericidal effect through medium acidification, glucose depletion, or the activity of antimicrobial peptides (bacteriocins) [30]. Indeed, *C. maltaromaticum* was the only isolate capable of inhibiting *L. monocytogenes* (<0.2 log CFU g^−1^ at 24 h). Strains vary in their ability to produce bacteriocins [31]. So far, studies have shown *L. monocytogenes* inhibition by *C. divergens* V41 (due to divercin V41) in smoked salmon medium [32], and *C. maltaromaticum* C2 (due to bacteriocins) in fish broth medium [33]. *C. divergens* M35 is already approved as an antimicrobial additive in cold-smoked salmon and trout in Canada [34]. While inoculation of *Carnobacterium* strains does not result in major sensory changes in cold-smoked salmon [35], in vitro studies of other strains need further evidence of antilisterial and antimicrobial activity in the product [36]. Conversely, other bacterial species have stimulated the growth of *L. monocytogenes*, which could be attributed to facilitated access to nutrients (e.g., amino acids) as a result of other species’ activity (e.g., the release of proteases) [37].

Another strategy tested to improve the shelf-life and microbiological safety of fresh salmon was using seasonings as biopreservatives. Concentrations of seasonings inhibiting *L. monocytogenes* were also inhibitory for spoilage bacteria. *L. monocytogenes* strains were not inhibited by NaCl (MIC ≥ 10%), as expected, due to multiple osmoprotective mechanisms, including salt shock proteins and cytoplasmic accumulation of osmoprotectants [38]. Conversely, in vitro MIC of both *L. monocytogenes* strains was 0.13% for OEO, 10% for lemon juice (i.e., ~0.5% of citric acid considering a 5% concentration), and 50 mg mL^−1^ for garlic powder. This agrees with previous reports of MIC for *L. monocytogenes* which were found to be 0.20% for OEO, 0.63% for citric acid, and 0.40% for garlic oil [15]. Indeed, OEO shows the greatest potential as a biopreservative. OEO preserves the flavor, odor, and texture of fresh salmon [15] while presenting multiple active substances (e.g., carvacrol) with antimicrobial activity by damaging cell structures (e.g., membrane, cell wall) and impairing proton gradients [39]. Moreover, Mixture 1 (5% NaCl, 0.002% OEO, 2.5% lemon juice, 0.08 mg mL^−1^ garlic powder) was the best-performing seasoning, reducing the number of total microorganisms and *Pseudomonas* spp. by 7–8 log CFU g^−1^ at 16 h (at <2 log CFU g^−1^) and completely inhibiting Gram-positive bacteria and *Listeria* spp.

The current exploratory work has identified three areas relevant for increasing fresh salmon shelf-life: (i) good hygiene practices to reduce initial microbial contamination; (ii) vacuum packaging under refrigeration (4 °C); and (iii) use of biopreservatives, individually (i.e., 0.13% OEO) or in combination (i.e., Mixture 1). While these strategies were successful in vitro, they need to be further explored in fish culture models and, most importantly, directly in fish fillets while also considering sensory changes which may deter consumers. The combination of vacuum packaging and inoculation of *Lactococcus lactis* spp. *lactis* has been shown to extend the shelf-life of fresh salmon without sensory changes [40]. Therefore, future works should study the effects of a combination of the strategies identified in the present study on spoilage, food safety, and sensory changes of fresh salmon.

A good correlation (r_s_ ≥ 0.827) was found for the 10 isolates between the commercial culture medium and the salmon medium (prepared from macerated salmon fillets). The salmon medium showed a significant reduction in the maximum population density of 0.6 log, possibly related to the presence of antimicrobial peptides [41]. Previous works have also used salmon flesh medium to test the proteolytic and lipolytic activity and to identify spoilage bacteria (those producing clear zones from hydrolytic activity) [42], and fish flesh medium to test the antimicrobial properties of ceviche components [43]. Therefore, follow-up studies may use the salmon medium as a more realistic condition before tests on salmon fillets.

## 5. Conclusions

This work aimed at identifying potential strategies to improve the shelf-life and microbiological safety of fresh salmon. Storage of fresh salmon fillets under vacuum at 4 °C extended shelf-life, based on acceptability criteria, up to 20 days, and reduced microbial growth by 3.3 log CFU g^−1^. Moreover, this was only possible due to a lower initial flora (0–1 log CFU g^−1^) likely stemming from good hygiene and chilling practices. *C. maltaromaticum* thrived under vacuum storage and was the only isolate capable of inhibiting *L. monocytogenes* (<0.2 log CFU g^−1^ at 24 h), likely due to the release of antimicrobial substances. Therefore, *C. maltaromaticum* has the potential to be an antimicrobial additive for fish products. *L. monocytogenes* generally required higher seasoning concentrations for inhibition when compared to spoilage bacteria. OEO (0.13%) individually or a mixture of seasoning (5% NaCl, 0.002% OEO, 2.5% lemon juice, 0.08 mg mL^−1^ garlic powder) produced good inhibitory activity in vitro, including for *L. monocytogenes*. Therefore, fresh salmon shelf-life can be readily improved by good hygiene practices and vacuum packaging, while *C. maltaromaticum,* OEO, and seasoning mixtures are promising alternatives that require further testing.

## Figures and Tables

**Figure 1 vetsci-10-00423-f001:**
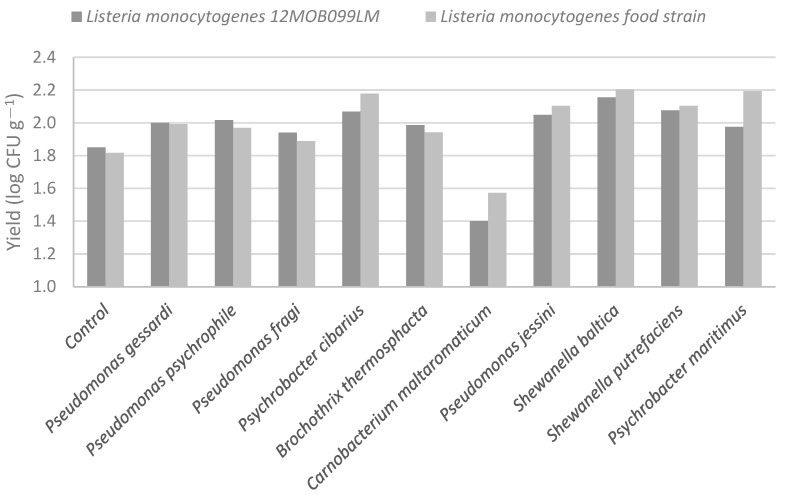
Growth of *Listeria monocytogenes* strains in the presence of spoilage bacteria isolated from salmon fillets.

**Figure 2 vetsci-10-00423-f002:**
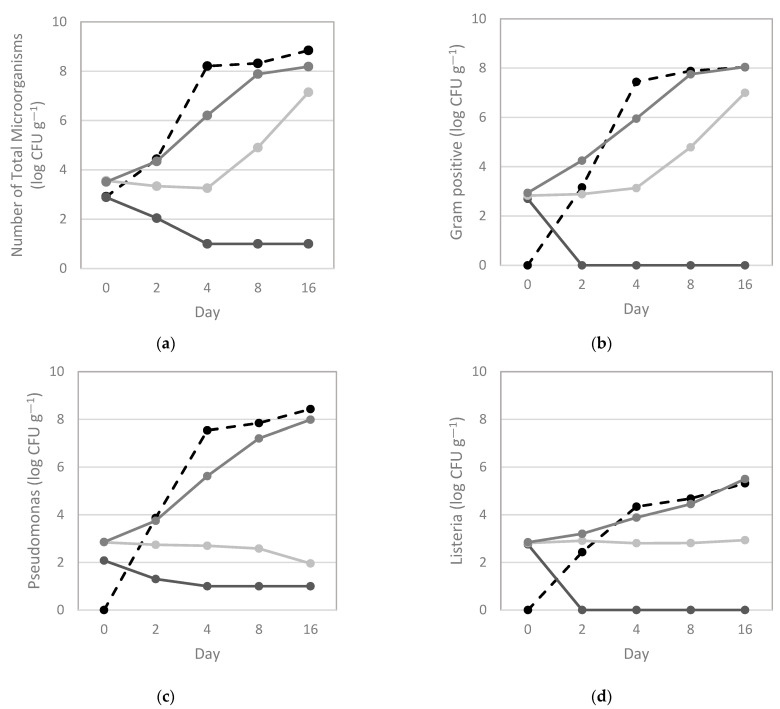
Effects of seasoning mixtures on the counts of (**a**) total microorganisms, (**b**) Gram-positive bacteria, (**c**) *Pseudomonas* spp., and (**d**) *Listeria* spp. (log colony forming units (CFU) g^−1^) at 0, 2, 4, 8, and 16 days of storage at 4 °C. Mixture 1 (━) contained 5% NaCl, 0.002% OEO, 2.5% lemon juice, and 0.08 mg mL^−1^ garlic powder, encompassing at least one MIC value for the 10 representative bacterial species found during storage. Mixture 2 (━) was based on the MIC values of the *Listeria monocytogenes* strains and was formulated with 2.5% NaCl, 0.002% OEC, 0.625% lemon juice, and 3.13 mg mL^−1^ garlic powder. Mixture 3 (━), with 0.25% NaCl, 0.005% OEO, 0.25% lemon juice, and 0.06 mg mL^−1^ garlic powder, was based on the concentration expected from the absorption marinades used in the industry. Control (**┅**).

**Table 1 vetsci-10-00423-t001:** In vitro activities of four natural preservatives against two *Listeria monocytogenes* strains and 10 bacterial species selected among spoilage bacteria of salmon fillets stored under air and commercial vacuum packaging, expressed as the minimal inhibitory concentration (MIC) and the minimal bactericidal concentration (MBC).

Bacterial Species	NaCl (%)		Oregano Essential Oil (%)		Lemon Juice (%)		Garlic Powder (mg mL^−1^)	
	MIC	MBC	MIC	MBC	MIC	MBC	MIC	MBC
*Pseudomonas gessardi*	5	10	0.03	>1	2.5	2.5	12.5	25
*Pseudomonas psychrophile*	5	10	0.03	0.5	2.5	5	12.5	12.5
*Pseudomonas fragi*	5	10	0.03	>1	2.5	5	6.25	12.5
*Psychrobacter cibarius*	10	10	0.004	0.004	0.625	1.25	3.13	12.5
*Brochothrix thermosphacta*	10	10	0.016	0.03	2.5	2.5	3.13	3.13
*Carnobacterium maltaromaticum*	10	10	0.016	0.3	2.5	2.5	12.5	3.13
*Pseudomonas jessini*	5	10	0.016	>1	2.5	2.5	12.5	25
*Shewanella baltica*	5	>10	0.004	0.008	1.25	2.5	6.25	12.5
*Shewanella putrefaciens*	5	>10	0.004	0.008	1.25	2.5	6.25	12.5
*Psychrobacter maritimus*	5	>10	0.004	0.016	1.25	2.5	6.25	12.5
*Listeria monocytogenes* 12MOB099LM	10	>10	0.13	0.13	5	10	12.5	25
*Listeria monocytogenes* (food strain)	>10	>10	0.06	0.25	2.5	10	12.5	50

## Data Availability

Data are available upon request.

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
