# Peer review of "An Exploratory Study on Spoilage Bacteria and Listeria monocytogenes in Fresh Salmon: Extending Shelf-Life Using Vacuum and Seasonings as Natural Preservatives"

_vetsci, 2023, doi:10.3390/vetsci10070423_

Round 1

Reviewer 1 Report

Review: „ Exploratory Study on Spoilage Bacteria and Listeria monocytogenes in Fresh Salmon: Extending Shelf-Life Using Vacuum and Seasoning as Natural Preservatives”

The authors describe experiments where they investigating the impact of vacuum packing, distinct seasoning, and the implementation of bactericidal microorganisms. The aim of the study is the development of new, holistic strategies to combat spoiling microorganisms. The manuscript is well written but needs some further spelling verification, which I cannot provide. I have some concerns about the relevance of the manuscript. As mentioned shortly before, the manuscript focuses on new strategies for combatting spoiling microorganisms. However, all aspects mentioned alone only merely represent any novelty. The authors should rather focus on the whole strategy embracing all aspects of their experiments and how they can be merged to a single strategy. Guide the reader more strongly throughout your manuscript. Explain your findings against the background of a holistic strategy, introduce your hypotheses and discus them against the background of your findings. In the text at hand, a global strategy is mentioned only in the conclusions.

General remarks:

1)      Please introduce the kind of logarithm you are using (If it is decadic, either log10 or lg)

2)      Please describe your figures more carefully. Show the reader the statistics (arithmetic mean, error bars, etc.) you made and which is mentioned in the materials and methods.  

3)      You talking about significant results. How do you judge significance?

4)      Describe multiple layers on unified images in a greater detail, such as Figure 2. (Call it fig 2 a or fig. 2 b, etc.)

5)      The discussion left some points open and unmentioned. Especially against the background of a wider strategy for combating microorganisms.

The manuscript is suitable for publishing in “veterinary sciences” after major revisions.

Author Response

We would like to thank the editor and reviewers for the time they have spent on improving our manuscript and suggesting valuable corrections. We have analyzed and addressed them carefully. These changes and our responses are indicated below in blue and as tracked changes in the manuscript. Hopefully, this manuscript is now suitable for publication.

Reviewer 1:

The authors describe experiments where they investigating the impact of vacuum packing, distinct seasoning, and the implementation of bactericidal microorganisms. The aim of the study is the development of new, holistic strategies to combat spoiling microorganisms. The manuscript is well written but needs some further spelling verification, which I cannot provide.

R: The manuscript has been revised by a C2 level English speaker and, additionally, using a grammar correction software (Grammarly).

I have some concerns about the relevance of the manuscript. As mentioned shortly before, the manuscript focuses on new strategies for combatting spoiling microorganisms. However, all aspects mentioned alone only merely represent any novelty. The authors should rather focus on the whole strategy embracing all aspects of their experiments and how they can be merged to a single strategy. Guide the reader more strongly throughout your manuscript. Explain your findings against the background of a holistic strategy, introduce your hypotheses and discuss them against the background of your findings. In the text at hand, a global strategy is mentioned only in the conclusions.

R: Thank you for your suggestion. Indeed, our purpose is to identify strategies that can be further explored in more detailed studies. According to the suggestion, we have made changes to the introduction and discussion.

Introduction: we have improved the last part of the introduction to better show the novelty of the work, namely the use of seasonings to extend the shelf-life of fresh salmon (while most literature is on smoked salmon) and reorganized the objective to be more easily comprehensible.

“Therefore, fresh salmon is a main candidate for biopreservation, that is, preservation using natural microorganisms and antimicrobials. So far, limited research has been conducted on fresh salmon preservation compared to smoked salmon, especially addressing novel preservation strategies (e.g., biopreservatives). Thus, this exploratory work aimed at identifying possible strategies to extend the shelf-life and microbiological safety of fresh salmon, measuring the capacity for microbial inhibition of an available and well-recognized method (vacuum packaging) and of an emerging strategy using natural preservatives (seasonings).”

Discussion: To emphasize this aspect, we have revised the following paragraph of the discussion to include reference to the combination of different strategies, as suggested.

“The current exploratory work has identified three areas relevant for increasing fresh salmon shelf-life: i) good hygiene practices to reduce initial microbial contamination; ii) vacuum packaging under refrigeration (4 ˚C); and iii) use of biopreservatives, individually (i.e., 0.13% OEO) or in combination (i.e., Mixture 1). While these strategies were successful in vitro, they need to be further explored in fish culture models and, most importantly, directly in fish fillets while also considering sensory changes which may deter consumers. The combination of vacuum packaging and inoculation of Lactococcus lactis spp. lactis has been shown to extend the shelf-life of fresh salmon without sensory changes [41]. Therefore, future works should study the effects of a combination of the strategies identified in the present study on spoilage, food safety, and sensory changes of fresh salmon.” 

  • Please introduce the kind of logarithm you are using (If it is decadic, either log10or lg)

R: Thank you for this suggestion. We have added the following sentence on the subsection “2.5 Statistical Analysis”:

“Results are expressed as base 10 logarithm (log).”

  • Please describe your figures more carefully. Show the reader the statistics (arithmetic mean, error bars, etc.) you made and which is mentioned in the materials and methods.  

R: Thank you for the suggestion. All statistics that can be conducted (and are mentioned in Materials and Methods) are already detailed in the manuscript, including the p-value and the average or median (depending on the test conducted) for each treatment. You can find this, for instance, on section 3.3. (e.g., “p = 0.463; 0.745 vs.0.740”). We used this carefully as introducing a large list of numbers in text could make it difficult to read. Therefore, we have spaced them withing the corresponding sentences (as interpretation) and using only the necessary numbers, to not distract the reader.

  • You talking about significant results. How do you judge significance?

R: The two uses of “significant” in text include i) results from the present study supported by the p-value, ii) a literature reference which the authors found a statistically significant reduction. In the present study, significance is judged by a statistical criterion, namely, α=0.05 (as established in subsection 2.5).

“Statistically significant differences were found for the maximum population density, with a reduction of 0.6 log cycle in the salmon media (p = 0.046; 9.180 vs. 8.475 log CFU g-1).”

“The salmon medium showed a significant reduction in the maximum population density of 0.6 log, possibly related to the presence of antimicrobial peptides [40].”

  • Describe multiple layers on unified images in a greater detail, such as Figure 2. (Call it fig 2 a or fig. 2 b, etc.)

R: This is a good point. We have identified each panel (a – d) and revised to text describing the results to include it, to improve the description of the figure.

“Based on MIC results, three seasoning mixtures were prepared using different concentrations (Figure 2). Mixture 1 inhibited total aerobic microorganisms at 28 ˚C during the full test period, corresponding to a 7.9 log CFU g-1 reduction at 16 h (Figure 2, a). Mixture 2 and Mixture 3 had lower inhibitory activities (1.70 and 0.66 log CFU g-1 at 16 h, respectively), which likely results from the lower concentrations of NaCl and lemon when compared to Mixture 1 (Figure 2, a). Regarding specific groups of bacteria, Gram-positive bacteria presented an inhibition trend similar to the number of total microorganisms but with complete inhibition by Mixture 1 (Figure 2, b). Pseudomonas spp. was inhibited by Mixture 1 and Mixture 2 (reduction of 7.4 and 6.5 log CFU g-1 at 16 h, respectively; Figure 2, c) and both L. monocytogenes strains were completely inhibited by Mixture 1 (Figure 2, d).”

  • The discussion left some points open and unmentioned. Especially against the background of a wider strategy for combating microorganisms.

R: Thank you for this suggestion. As previously mentioned, we have addressed this aspect by rewriting the following section of the discussion: 

“The current exploratory work has identified three areas relevant for increasing fresh salmon shelf-life: i) good hygiene practices to reduce initial microbial contamination; ii) vacuum packaging under refrigeration (4 ˚C); and iii) use of biopreservatives, individually (i.e., 0.13% OEO) or in combination (i.e., Mixture 1). While these strategies were successful in vitro, they need to be further explored in fish culture models and, most importantly, directly in fish fillets while also considering sensory changes which may deter consumers. The combination of vacuum packaging and inoculation of Lactococcus lactis spp. lactis has been shown to extend the shelf-life of fresh salmon without sensory changes [41]. Therefore, future works should study the effects of a combination of the strategies identified in the present study on spoilage, food safety, and sensory changes of fresh salmon.” 

The manuscript is suitable for publishing in “veterinary sciences” after major revisions.

R: We are grateful for the reviewer’s comments that have helped us improve this manuscript.

Reviewer 2 Report

This study concerns the in vitro interactions of spoilage bacteria and Listeria monocytogenes isolated from salmon. The title as it appears now is misleading as it gives the impression that these interactions were studied in a food model.

The in vitro interaction studies as well as inhibition studies were conducted at 28 C which is not a realistic scenario for this product that was stored at 4 C. I miss the justification for the high temperature used and the autors need to reflect if such a set-up says something meaningful in relation to storage at 4 C.

The coc-culture studies indicate that growth of L. monocytogenes is increased in presence of all isolates except C. maltaromaticum. This is a surprising result (and contradict the Jameson effect) which needs to be discussed.

It is very simple to test if the inhibitory effect by the C. maltaromaticum culture is due to bacteriocinogenic activity by employing a deferred inhibition agar assay and adding proteinase K next to the producer culture before overlay with the target culture.

Specific comments:

1) l. 122-123: e.g. "selection criteria" instead of "tiebreaker"

2) line 171-192: Please, add incubation conditions used for the different media.

3) line 217: "aerobic" instead of "aerobiotic"

4) Figure 2: Explanation of symbols is missing

5)  Appendix A: Please reorganize so that genera and species are arranged alphabetically

6) Ref 35 should be modified according to the requirements by the journal.

In general the language is fine, however it would be suitable to check it carefully by a reader proficient in English.

Author Response

This study concerns the in vitro interactions of spoilage bacteria and Listeria monocytogenes isolated from salmon. The title as it appears now is misleading as it gives the impression that these interactions were studied in a food model.

R: The study has indeed been conducted on a food model. The first part of the work pertained to the study of fresh salmon fillets and the isolation of bacterial dynamics in that food model. Additional tests were conducted in vitro based on the species previously isolated, as described in the first section. Moreover, we have conducted a test on the use of fish fillet model medium which showed a high correlation to commercial medium, supporting in vitro results using commercial media. We have often emphasized what kind of tests we were referring to in section titles and in tests. For instance, as “3.2. In Vitro Antimicrobial Effect of Different Natural Preservatives (Seasonings) and Mixtures”.

The in vitro interaction studies as well as inhibition studies were conducted at 28 C which is not a realistic scenario for this product that was stored at 4 C. I miss the justification for the high temperature used and the autors need to reflect if such a set-up says something meaningful in relation to storage at 4 C.

R: We understand the reviewer’s concern. However, we would like to emphasize that 28ºC refers to the incubation of isolated microorganisms, which follows well-established international protocols detailed in the materials and methods. The guidelines developed by CLSI recommend an incubation temperature of 28°C for bacteria isolated from aquatic animals for broth dilutions assays. The incubation of the fresh salmon in the first test was conducted at 4ºC, a realistic scenario just as pointed out by the reviewer.

The co-culture studies indicate that growth of L. monocytogenes is increased in presence of all isolates except C. maltaromaticum. This is a surprising result (and contradict the Jameson effect) which needs to be discussed.

R: Thank you for this suggestion. We have indeed focused mostly on the effect that could be exploited to expand shelf-life (following the objective of the work). Nonetheless, we agree that the growth stimulation of Listeria is also worth mentioning in the discussion. Therefore, we have added the following sentence:

“Conversely, other bacterial species have stimulated the growth of L. monocytogenes, which could be attributed to a facilitated access to nutrients (e.g., amino acids) as a result of other species’ activity (e.g., release of proteases) [38].”

It is very simple to test if the inhibitory effect by the C. maltaromaticum culture is due to bacteriocinogenic activity by employing a deferred inhibition agar assay and adding proteinase K next to the producer culture before overlay with the target culture.

R: As we have discussed in the manuscript, inhibition may be due to antimicrobial peptides, acidification, or nutrient depletion. Your recommendation of using the deferred inhibition agar assays with proteinase K will come in handy when we study these hypotheses in more detail in a future study. Thank you.

1) l. 122-123: e.g. "selection criteria" instead of "tiebreaker"

R: We have changed to “tiebreaker” has suggested.

2) line 171-192: Please, add incubation conditions used for the different media.

R: The incubation conditions were already described in a previous sentence of the same paragraph.

“The bacterial suspensions (set of bacteria with each mixture) were then incubated at 28 °C.” Nonetheless, we have added that information when referring to the selective media as suggested: “Additionally, 100 µl of the bacterial suspensions were also inoculated on six selective me-dia in order to distinguish bacterial species and incubated at 28 ˚C.”

3) line 217: "aerobic" instead of "aerobiotic"

R: Corrected. Thank you.

4) Figure 2: Explanation of symbols is missing

R: Thank you for noticing this typo. We have included this in a previous version, but it was lost during formatting. We have now added this information again to the present manuscript.

Figure 2. Effect of seasoning mixtures on the counts of total microrganisms, Gram-positive bacteria, Pseudomonas spp. and Listeria spp. (log colony forming units (CFU) g−1) at 0, 2, 4, 8  and 16 days of storage at 4˚C. Mixture 1 (━) contained 5% NaCl, 0.002% OEO, 2.5% lemon juice, and 0.08 mg mL-1 garlic powder, encompassing at least one MIC value for the 10 representative bacterial species found during storage. Mixture 2 (━) was based on the MIC values of the Listeria monocytogenes strains and was formulated with 2.5% NaCl, 0.002% OEC, 0.625% lemon juice, 3.13 mg mL-1 garlic powder. Mixture 3 (━),with 0.25% NaCl, 0.005% OEO, 0.25% lemon juice, 0.06 mg mL-1 garlic powder, was based on the concentration expected from the absorption marinades used in the industry. Control ()”

5)  Appendix A: Please reorganize so that genera and species are arranged alphabetically

R: Thank you for this suggestion. We have rearranged the table according to instructions.

Isolate

Bacterial species

Isolate

Bacterial species

E2M1P/V2

Arthrobacter alpinus

E2M3/A1

Pseudomonas fragi

E1M2P/V6

Brochothrix thermosphacta

E2M5/A1

Pseudomonas fragi

M5AN2

Brochothrix thermosphacta

E2M5P/A1

Pseudomonas fragi

E2M2/A4

Brochothrix thermosphacta

E2M1P/V4

Pseudomonas fragi

E2M1/V2

Buttiauxella gaviniae

E1M1/A1

Pseudomonas gessardi

E1M1/V2

Carnobacterium maltaromaticum

E1M1/A4

Pseudomonas gessardi

E1M2/V1

Carnobacterium maltaromaticum

E1M1P/A2

Pseudomonas gessardi

E1M2/V4

Carnobacterium maltaromaticum

E1M2/A1

Pseudomonas gessardi

E1M4A/V2

Carnobacterium maltaromaticum

E1M1/V4

Pseudomonas jessinii

M5VA2

Carnobacterium maltaromaticum

E1M3/V1

Pseudomonas jessinii

M5AN3

Carnobacterium maltaromaticum

E2M4/A3

Pseudomonas poae/P. trivialis

E2M1/V1

Carnobacterium maltaromaticum

E1M2/A2

Pseudomonas psychrophila

E2M2/V1

Carnobacterium maltaromaticum

E1M3P/A2

Pseudomonas psychrophila

E2M2/V3

Carnobacterium maltaromaticum

E2M2/A1

Pseudomonas psychrophila

E2M2P/V1

Carnobacterium maltaromaticum

E2M2P/A1

Pseudomonas versuta/P. fragi

E2M3/V2

Carnobacterium maltaromaticum

E1M4/V3

Pseudomonas weihenstephanensis

E2M5/V3

Carnobacterium maltaromaticum

E1M3P/V3

Psychrobacter cibarius

E2M6/V1

Carnobacterium maltaromaticum

E2M4P/A2

Psychrobacter cibarius

E1M1/V1

Escherichia coli

E2M1P/A1

Psychrobacter cibarius

E2M4/V1

Moraxella osloensis

E2M4P/V3

Psychrobacter maritimus

E2M3/V1

Paenibacillus cineris/P. rhizosphaerae

E2M1/A5

Psychrobacter nivimaris

E2M1/A4

Pseudomonas antarctica

E2M5P/V2

Psychrobacter sp.

E1M1P/A3

Pseudomonas fragi

E1M1P/V2

Shewanella baltica

E1M2P/A1

Pseudomonas fragi

E1M1P/V3

Shewanella baltica

M5VA4

Pseudomonas fragi

M5AE1

Shewanella putrefaciens

6) Ref 35 should be modified according to the requirements by the journal.

R: This has been corrected. Thanks.

Round 2

Reviewer 2 Report

The manuscript has now been improved.